

# A progressive attention-based cross-modal fusion network for cardiovascular disease detection using synchronized electrocardiogram and phonocardiogram signals

Wei Peng Li[1], Joon Huang Chuah[1], Guo Jeng Tan[2], Chengyu Liu[3] and Hua-Nong Ting[4,5]

[1] Department of Electrical Engineering, Faculty of Engineering, Universiti Malaya, Kuala Lumpur, Kuala Lumpur, Malaysia
[2] Department of Medicine, Faculty of Medicine, Universiti Malaya, Kuala Lumpur, Kuala Lumpur, Malaysia
[3] School of Instrument Science and Engineering, Southeast University, Nanjing, JiangSu, China
[4] Department of Biomedical Engineering, Faculty of Engineering, Universiti Malaya, Kuala Lumpur, Kuala Lumpur, Malaysia
[5] School of Medical Engineering, Jining Medical University, Jining City, Shandong Province, China

Corresponding author
Hua-Nong Ting, tinghn@um.edu.my

## ABSTRACT

Synchronized electrocardiogram (ECG) and phonocardiogram (PCG) signals provide complementary diagnostic insights crucial for improving the accuracy of cardiovascular disease (CVD) detection. However, existing deep learning methods often utilize single-modal data or employ simplistic early or late fusion strategies, which inadequately capture the complex, hierarchical interdependencies between these modalities, thereby limiting detection performance. This study introduces PACFNet, a novel progressive attention-based cross-modal feature fusion network, for end-to-end CVD detection. PACFNet features a three-branch architecture: two modality-specific encoders for ECG and PCG, and a progressive selective attention-based cross-modal fusion encoder. A key innovation is its four-layer progressive fusion mechanism, which integrates multi-modal information from low-level morphological details to high-level semantic representations. This is achieved by selective attention-based cross-modal fusion (SACMF) modules at each progressive level, employing cascaded spatial and channel attention to dynamically emphasize salient feature contributions across modalities, thus significantly enhancing feature learning. Signals are pre-processed using a beat-to-beat segmentation approach to analyze individual cardiac cycles. Experimental validation on the public PhysioNet 2016 dataset demonstrates PACFNet's state-of-the-art performance, with an accuracy of 97.7%, sensitivity of 98%, specificity of 97.3%, and an F1-score of 99.7%. Notably, PACFNet not only excels in multi-modal settings but also maintains robust diagnostic capabilities even with missing modalities, underscoring its practical effectiveness and reliability. The source code is publicly available on Zenodo (https://zenodo.org/records/15450169).

# INTRODUCTION

Cardiovascular diseases (CVDs) are a major concern for global health. They encompass conditions affecting the heart and blood vessels, such as coronary heart disease, heart failure, arrhythmias, and hypertension (*Townsend et al., 2022*). CVDs are a leading cause of mortality worldwide. The World Health Organization (WHO) reported approximately 17.9 million deaths from CVDs in 2019, representing 32% of all global deaths (*World Health Organization, 2021*).

Current CVD diagnosis relies on several methods. These include phonocardiography (PCG), electrocardiography (ECG), echocardiography, and coronary angiography. Among these, ECG and PCG are frequently used for initial CVD diagnosis. Their advantages include non-invasiveness, rapid results, and cost-effectiveness. ECG records the heart's electrical activity, identifying waveform pattern changes to diagnose various heart diseases (*Jahmunah et al., 2019*; *Li et al., 2022a*). PCG records heart sounds, detecting abnormal valve function or structural cardiac issues (*Zhu et al., 2024*). Combined ECG and PCG signals provide comprehensive information which could capture both the electrical and mechanical aspects of cardiac function. Consequently, diagnostic accuracy for CVDs is improved. This is particularly beneficial in identifying at-risk patients who may not exhibit obvious symptoms.

The conventional clinical diagnosis of CVDs relies significantly on the interpretation of ECG and PCG signals by physicians. However, this approach has inherent limitations: it is time-consuming (*Xu, Mak & Chang, 2022*), potentially delaying critical interventions, and its accuracy is heavily dependent on extensive physician experience and specialized skills (*Jiang & Choi, 2006*), introducing subjectivity and inter-observer variability. Furthermore, the resource-intensive nature of training proficient cardiologists, both temporally and economically, exacerbates the scarcity of expert personnel, an issue particularly acute in less developed regions (*Hu et al., 2024*). The recent proliferation of portable and wearable devices for out-of-hospital ECG and PCG monitoring, while promising for continuous health surveillance, introduces a new challenge, as the sheer volume of data often overwhelms the capacity for real-time physician review (*Emmett et al., 2023*). Combined, these limitations underscore an urgent and unmet clinical need for an automated, objective, and accurate methodology capable of diagnosing CVDs. Therefore, the primary objective of this research is to develop and validate a novel computational framework that leverages the complementary diagnostic information of multimodal signals from ECG and PCG. This framework aims to provide a robust, efficient, and reliable diagnostic tool, thereby reducing the burden on healthcare professionals, and improving accessibility to cardiovascular diagnostics. The development of such intelligent algorithms for processing

and interpreting ECG and PCG signals has consequently become a significant focus of current research.

## Literature review

Recent research has explored automated diagnosis of CVD and other diseases using ECG and PCG signals (*Ameen et al., 2024*; *Huang et al., 2022*; *Allegra et al., 2023*; *Tasci et al., 2024*). These studies can be broadly categorized into two approaches: manual feature extraction and end-to-end feature extraction using deep learning. The manual feature extraction approach typically involves two steps. First, morphological and time-frequency domain features are extracted from the input signals. Second, these features are classified using machine learning or deep learning methods (*Chakir et al., 2020*). For instance, *Singh et al. (2021)* extracted over ten time-frequency domain statistical features from synchronized ECG and PCG signals. The authors compared the performance of multiple classifiers. A support vector machine (SVM) classifier achieved the highest accuracy of 93.1%. *Li et al. (2022b)* used separate SVM classifiers for ECG and PCG signal branches. A Dempster-Shafer (D-S) theory-based strategy fused the classification results from both modalities, achieving a final accuracy of 86.4%. *Jyothi & Pradeepini (2024)* decomposed ECG and PCG signals using the improved empirical mode decomposition (IEMD) algorithm, extracting morphological and time-frequency domain features. Following optimization and feature selection with the I-CSOA algorithm, these features were concatenated pairwise and classified using the Gaussian Kaiming variance-based deep learning neural network (GKVDLNN), categorizing signals into Normal, Arrhythmia, Mitral Valve Prolapse, Ischemia, and Valvular Heart Disease. Nevertheless, methods relying on manual feature extraction and classification often face limitations. These include insufficient learning of modal features, potential omission of important features, and limited generalizability and robustness.

In contrast to manual feature extraction, deep learning models offer several advantages. They eliminate the need for hand-designed features, can also discover complex patterns that are difficult for humans to discern (*Liu et al., 2023*; *Bhardwaj, Singh & Joshi, 2023*). Consequently, they are increasingly employed for automatic classification of multimodal ECG and PCG signals. Existing deep learning multimodal feature fusion strategies are typically categorized as early fusion, late fusion, or intermediate fusion (*Stahlschmidt, Ulfenborg & Synnergren, 2022*; *Boulahia et al., 2021*). Early fusion commonly employs a single-branch structure. In this approach, multimodal data are concatenated directly at the input stage. For example, *Ibrahim et al. (2024)* concatenated downsampled ECG and PCG signals. They then used a MobileNetV2 model for classification, achieving an accuracy of 97%. Despite this, this study trained the model five times on the same dataset without cross-validation. Therefore, the model's robustness requires further evaluation. *Li et al. (2019)* combined features from concatenated ECG and PCG signals with manually extracted multi-domain features, which were then used for classification. *Hangaragi et al. (2025)* concatenated ECG and PCG signals, then applied the Pan-Tompkins algorithm for

waveform extraction and peak detection. Subsequently, they employed the Heming Wayed Polar Bear Optimization algorithm for feature extraction and a C squared Pool Sign BI-power-activated deep convolutional neural network (DCNN) network for classification, which enabled effective multiclass classification of cardiovascular diseases.

Late fusion involves independent feature extraction for each modality. The extracted features, or the decision results from each modality, are subsequently fused. For instance, *Li et al. (2022c)* used a three-branch convolutional neural network (CNN) model. The inputs were the concatenated original ECG and PCG signals, the time-frequency maps of the ECG signals, and the time-frequency maps of the PCG signals. Decision-level fusion, using D-S theory, was performed after obtaining classification results from each branch. This achieved an accuracy of 96.1% and a specificity of 90.8%. *Li, Hu & Liu (2021)* used CNN models for separate feature extraction of ECG and PCG signals. A genetic algorithm fused the features from both branches, and an SVM classifier performed the final classification, achieving an accuracy of 94.4%. *Zhu et al. (2025)* proposed DDR-Net, training separate DDR-ECG-Net and DDR-PCG-Net versions for dedicated ECG and PCG feature extraction, respectively. The extracted modal features were then concatenated, and important features were selected using recursive feature elimination (RFE). An SVM classifier then processed these selected features, achieving 91.6% accuracy. In a different approach, *Kalatehjari et al. (2025)* employed a convolutional neural network-bidirectional long short-term memory (CNN-BiLSTM) model for independent feature extraction from ECG and PCG signals. The features from these two branches were then fused and classified using a fully connected layer incorporating a bilinear layer, obtaining 97% accuracy.

While some methods demonstrate good performance, both early and late fusion have remarkable limitations. Early fusion and late fusion may not fully utilize complementary information by only fusing low-level morphological features or high-level semantic features. Furthermore, these approaches often do not fully consider the relative contributions of feature vectors from different modalities. Fusion of multimodal branch decision results using D-S theory offers limited performance improvement (*Hao, Luo & Pan, 2021*).

Intermediate feature fusion utilizes a multi-branch structure, performing feature extraction on each modality separately. Crucially, it fuses the features from each branch during the feature extraction process. The fused features are then input into subsequent network layers for learning and classification. For example, *Qi et al. (2023)* employed the GADF algorithm to convert ECG and PCG signals into two-dimensional (2D) images. A Transformer model performed feature extraction and fusion of the two modalities. The fused features were then input into a down-sampling residual network for classification. Their study was able to achieve an accuracy of 94.3%. *Zhang et al. (2024)* proposed a multi-level feature extraction method for ECG and PCG signals. Feature fusion occurred concurrently with feature extraction at each level. A decision-level fusion strategy subsequently combined the decision results from two feature extraction branches and one feature fusion branch. This method achieved an accuracy of 94.4%. While the approach is

effective, it has a complex structure, a large number of parameters, and high computational resource demands.

## Motivation and contribution

Building upon the aforementioned research, we propose a progressive attention-based fusion network (PACFNet) for end-to-end CVD detection using synchronized ECG and PCG signals. This model employs an intermediate feature fusion strategy. Importantly, it is designed for both multimodal scenarios and maintains robust performance even with single-modality input. By segmenting ECG and PCG signals based on the cardiac cycle, PACFNet can accurately identify abnormal waveform characteristics, providing an effective approach for real-time cardiac anomaly detection. The salient contributions of this work are summarized below:

- We propose a novel cardiac state discrimination model that utilizes synchronized ECG and PCG signals as input. This model employs an intermediate fusion strategy to progressively extract features from superficial to deep levels and fuse them.
- Within the feature fusion module, we innovatively integrate features extracted from ECG and PCG signals with the fused features from the previous level using spatial and temporal attention mechanisms. This effectively evaluates the importance of each region within the cross-modal feature vectors.
- Synchronized ECG and PCG signals are segmented based on the cardiac cycle. This not only augments the dataset but also enables the model to more acutely identify the waveform characteristics of abnormal signals, facilitating real-time patient monitoring.
- The proposed model was evaluated on the PhysioNet/CinC Challenge 2016 dataset and compared with state-of-the-art methods. The results demonstrate that our proposed model outperforms existing models in terms of classification accuracy, sensitivity, specificity, and F1-score.

## METHODOLOGY

Figure 1 illustrates the overall framework for diagnosing cardiac status using synchronized ECG and PCG signals. The process begins with data preprocessing. ECG and PCG recordings are segmented into synchronized cardiac cycle segments based on provided annotations. These segments are then fed into PACFNet for features extraction and classification. Within PACFNet, ECG and PCG signal features undergo progressive fusion. Ultimately, the model outputs the predicted cardiac state category.

## The overall architecture of PACFNet

Figure 2 illustrates the overall architecture of our proposed PACFNet model, which employs a three-branch design comprising two identical modality-specific encoders (one for ECG, one for PCG) and a progressive feature fusion encoder. The ECG and PCG encoders receive synchronized ECG and PCG segments as input. They extract features at multiple levels, from superficial to deep, within each respective modality. Subsequently, the progressive feature fusion encoder systematically integrates these multi-level features

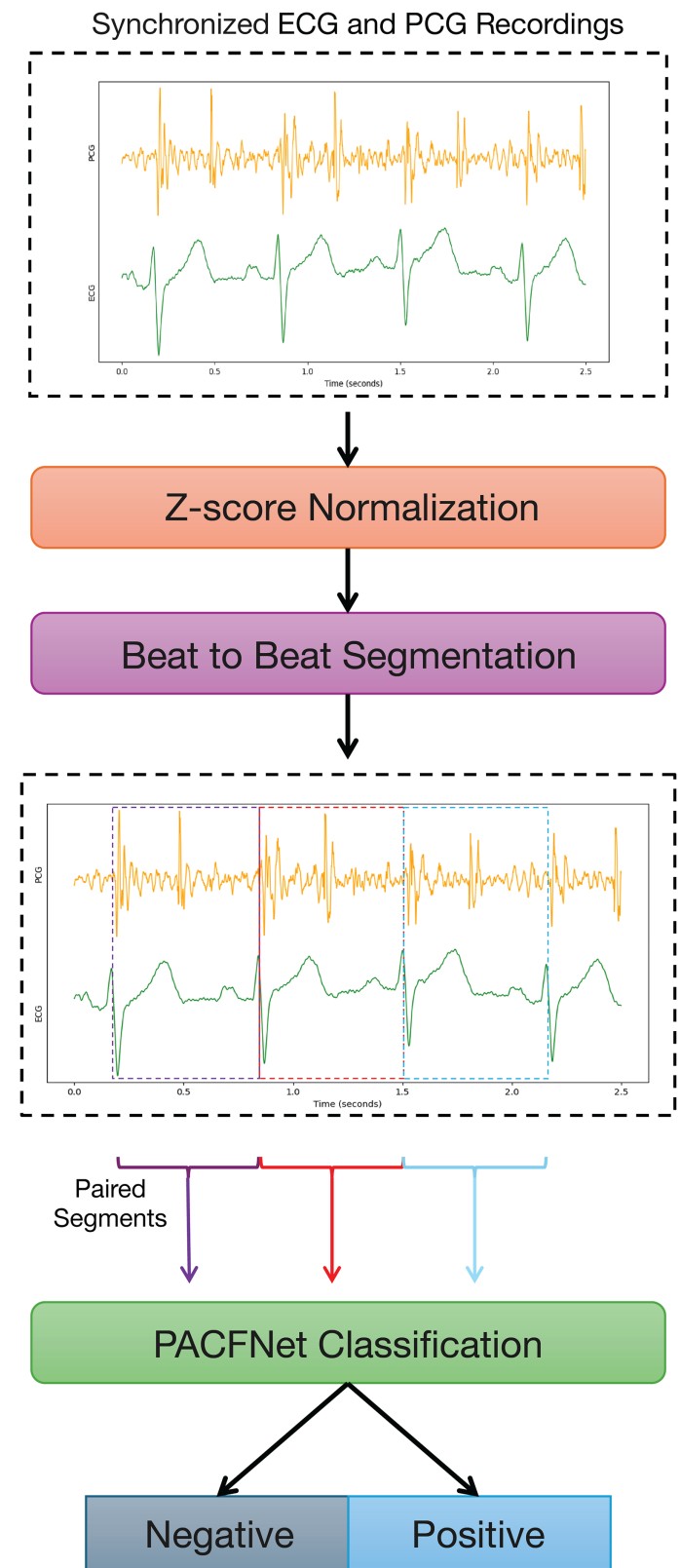

**Figure 1 Overall pipeline of the proposed multi-modality diagnosing framework for CVDs.**

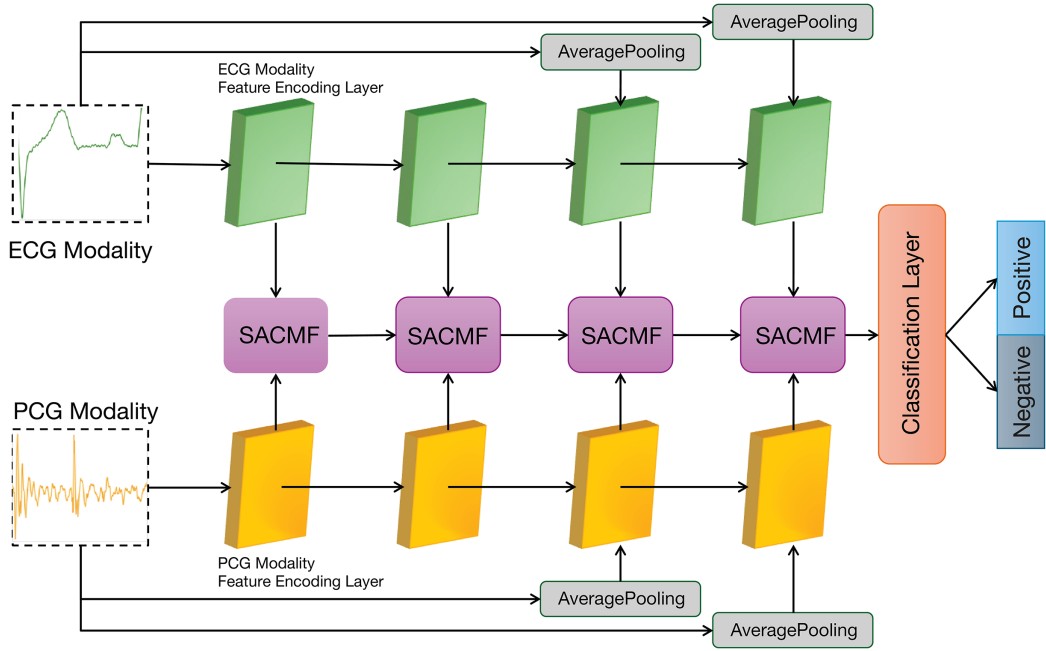

**Figure 2 Overall architecture of the PACFNet model.**

derived from both ECG and PCG. This integration is achieved through a series of selective attention-based cross-modal fusion (SACMF) modules that operate sequentially, progressing from shallower to deeper feature levels, with each SACMF module combining corresponding ECG and PCG features at its specific level. The resultant feature vector from the final SACMF module, representing the most deeply fused information, is then passed to the classification layer to produce the final cardiac state classification. By fusing ECG and PCG signals at multiple feature levels, the model leverages the complementary information present in both the electrical (ECG) and mechanical (PCG) activity of the heart. This approach enhances the accuracy and sensitivity of cardiac state recognition.

## The modal encoders for ECG and PCG signals

As shown in the Fig. 3, the proposed modal feature extraction module is inspired by the U-Net encoder architecture (*Ronneberger, Fischer & Brox, 2015*). The whole process comprises four identical feature extraction modules. Each module consists of an initial convolution-batch normalization-ReLU (CBR) block followed by two ResNet blocks connected in series.

Each feature extraction module progressively decreases the spatial dimensions of the input while concurrently enhancing the number of channels. This architectural strategy is designed to effectively capture increasingly abstract and contextual information from the signal. In the final two feature extraction modules, the original signal undergoes downsampling *via* average pooling (AP). This downsampled signal is then input into the feature extraction module, which enhances the representation of the original input signal's

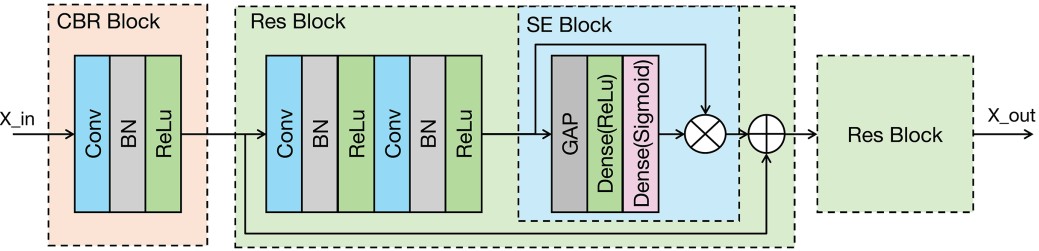

**Figure 3 Architecture of the proposed modality-specific feature extraction module.** Conv, Convolutional layer; ReLU, rectified linear unit; GAP, global average pooling; BN, batch normalization.

features. The feature extraction module within the modal encoder can be mathematically represented in Eq. (1).

$$f_i^M = \begin{cases} RES_{i_2}(RES_{i_1}(CBR(X))) & i \in \{1,2\} \\ AP(X_{in}) + RES_{i_2}(RES_{i_1}(CBR(X))) & i \in \{3,4\} \end{cases} \quad M \in \{ECG, PCG\} \tag{1}$$

where:

$i$ denotes the layer number of the feature extraction module.

$X$ is the input to the feature extraction module.

$X_{in}$ denotes the modal signal input at the very beginning.

$CBR(.)$ represents the CBR block operation.

$RES(.)$ represents the ResNet block operation.

$AP(.)$ represents the average pooling operation.

The CBR block performs an initial extraction of local signal features through a sequence of operations: convolution, batch normalization, and a ReLU activation function, where the ReLU activation enhances the model's capacity to learn complex, non-linear features. The ResNet block incorporates a residual connection (*He et al., 2016*), which facilitates the training of deeper networks. Importantly, the ResNet block in our model integrates a squeeze-and-excitation (SE) module (*Hu et al., 2020*). The SE module is a lightweight attention mechanism. It establishes interdependencies between feature channels and selectively enhances important feature channels while suppressing less relevant ones, which could improve the model's classification performance (*Jin et al., 2022*). The SE module primarily comprises two operations: squeeze and excitation. The output of the Excitation operation is then used to re-scale the input features of the SE block, performing channel-wise weighting. The SE module can be represented in Eq. (2).

$$\tilde{X} = X \odot \sigma(W_2 \delta(W_1 Fsq(X))) \tag{2}$$

where:

$X$ represents the input feature map.

$W_1, W_2$ are the weight matrix of the fully connected layer.

**Table 1 Detailed structural parameters of the modality-specific encoder.**

| Levels | Layers | Parameters | Layers | Parameters |
|--------|--------|------------|--------|------------|
| 1 | Conv-1 | C-64, K-7, S-1 | Conv-4 | C-64, K-7, S-1 |
|   | Conv-2 | C-64, K-7, S-1 | Conv-5 | C-64, K-7, S-1 |
|   | Conv-3 | C-64, K-7, S-1 |  |  |
| 2 | Conv-1 | C-128, K-7, S-5 | Conv-4 | C-128, K-7, S-5 |
|   | Conv-2 | C-128, K-7, S-5 | Conv-5 | C-128, K-7, S-5 |
|   | Conv-3 | C-128, K-7, S-5 |  |  |
| 3 | AP | 5 |  |  |
|   | Conv-1 | C-192, K-7, S-5 | Conv-4 | C-192, K-7, S-5 |
|   | Conv-2 | C-192, K-7, S-5 | Conv-5 | C-192, K-7, S-5 |
|   | Conv-3 | C-192, K-7, S-5 |  |  |
| 4 | AP | 25 |  |  |
|   | Conv-1 | C-256, K-7, S-5 | Conv-4 | C-256, K-7, S-5 |
|   | Conv-2 | C-256, K-7, S-5 | Conv-5 | C-256, K-7, S-5 |
|   | Conv-3 | C-256, K-7, S-5 |  |  |

Note:
AP represents an average pooling layer. C, K, S denote number of output channels, the kernel size, stride, respectively.

$\delta(.)$ and $\sigma(.)$ denote the ReLU and Sigmoid activation functions.

$Fsq(.)$ is the channel-wise global feature descriptor obtained *via* global average pooling (GAP).

$\odot$ represents element-wise multiplication along the channel dimension.

The structural parameters of the modality-specific encoders for ECG and PCG are detailed in Table 1. As the depth of feature extraction increases, the number of channels in the feature vectors also increases starting from 64. This allows the model to learn progressively higher-dimensional semantic features. The raw input signals are downsampled using average pooling layers and then fed into the third and fourth feature extraction modules. The pooling windows for these average pooling layers are 5 and 25, respectively.

### Selective attention-based cross-modal fusion module (SACMF)

The SACMF module is a critical component of PACFNet, designed to dynamically and adaptively determine the significance of information originating from different spatial locations and feature channels within the distinct ECG and PCG modal signals. This adaptive weighting allows the model to prioritize and select more discriminative features crucial for accurate cardiac state classification. Inspired by established attention mechanisms like the convolutional block attention module (CBAM) (*Woo et al., 2018*) and the method in *Zhang et al. (2024)* and *Roy, Navab & Wachinger (2019)*, the SACMF module sequentially computes attention maps along two independent dimensions: spatial and channel. These computed attention maps then act as modulators, being element-wise multiplied by the input feature vectors to perform adaptive feature modification, effectively recalibrating the feature representations. Figure 4 provides a detailed illustration of this

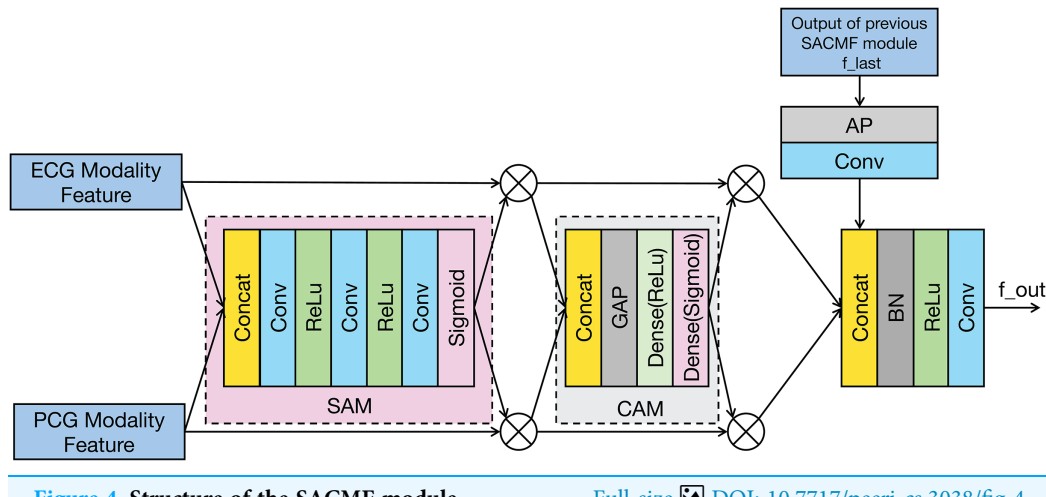

**Figure 4  Structure of the SACMF module.**  

cross-modal feature fusion process. As depicted, the operation of each SACMF module comprises three primary stages.

Spatial attention: At each progressive fusion level, feature vectors extracted from the corresponding ECG and PCG encoder layers are first concatenated. This combined multimodal feature map serves as input to a spatial attention module (SAM), whose objective is to identify 'where' the most salient information resides across the spatial dimensions by generating a attention weight map (*Woo et al., 2018*). This spatially-aware weight map is then element-wise multiplied by the concatenated feature vector, yielding a spatially-refined feature vector, denoted as $V_s$. This is accomplished through a sequence within the spatial attention weighting block: initially, a $1 \times 1$ convolution reduces channels in the concatenated multimodal feature vector to focus the subsequent spatial analysis; subsequently, a $16 \times 1$ convolution processes this reduced-channel map to explicitly learn spatial feature importance; and finally, to specifically address the potentially differing diagnostic regions of interest in ECG and PCG signals, a $1 \times 1$ convolution with two output channels, followed by a sigmoid activation function, generates two distinct, modality-specific spatial weight maps: one for ECG and one for PCG features. These maps highlight the critical spatial regions within each modality independently before they are applied to refine their respective feature contributions.

Channel attention: Following spatial refinement, the feature vector $V_s$ is passed to a channel attention module (CAM). Analogous to spatial attention identifying 'what' is important spatially, the CAM aims to determine 'which' feature channels are most informative (*Woo et al., 2018*). Structurally similar to the SE module, the CAM first applies global average pooling (GAP) to the input $V_s$. This operation aggregates spatial information to produce a channel descriptor, effectively summarizing the global context for each channel. Subsequently, this descriptor is fed through two fully connected (FC) layers—the first with a ReLU activation and the second with a sigmoid activation. These FC layers learn the non-linear interdependencies between channels and generate a channel-wise attention weight vector. This vector assigns an important score to each

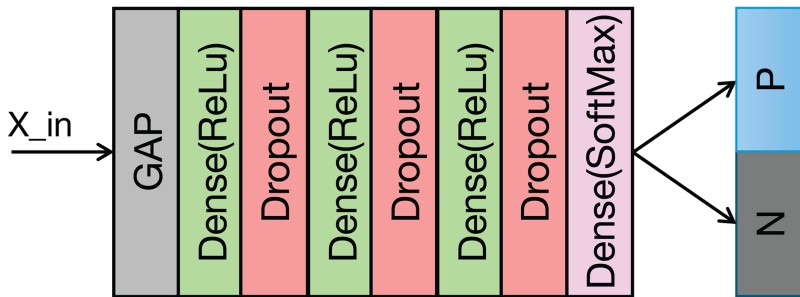

**Figure 5 Structure of the classification module.**

channel, which is then multiplied element-wise with $V_s$ to produce a channel-refined feature vector. This process selectively amplifies informative channels while attenuating less useful ones.

Fusion with Previous Level: The feature vector that has been adaptively refined by both spatial and channel attention mechanisms is fused with the multimodal fusion feature ($f_{last}$) propagated from the SACMF module of the preceding (shallower) fusion level. This integration is performed using a learnable convolution operation, allowing the model to combine the newly refined current-level features with the accumulated fused knowledge from earlier stages.

The cross-modal feature fusion module can be mathematically represented in Eq. (3).

$$\begin{cases} f_l = C_2\big(f_{l-1}^{last}, CAM(V_s)\big) \\ V_s = SAM\big(C_1\big(f_l^{ECG}, f_l^{PCG}\big)\big) \quad l \in \{1, 2, 3, 4\} \end{cases} \tag{3}$$

where:

   $f_l^{ECG}, f_l^{PCG}$ are the feature vectors of each level.
   $f_{l-1}^{last}$ represents the output from the previous SACMF module.
   $SAM(.)$ and $CAM(.)$ denote the SAM operation and Channel Attention Module operation.
   $C_1(.)$ is the Concatenation operation.
   $C_2(.)$ represents the fusion using batch normalization and convolution operations.

## The classification module

The structure of the classification module is illustrated in Fig. 5 and Table 2 details its structural parameters. The fused ECG and PCG features undergo downsampling *via* a convolutional operation. This reduces computational complexity and memory usage. Following downsampling, global average pooling (GAP) compresses the $L \times C$ feature vector to a $1 \times C$ vector. This approach provides two key advantages: it significantly reduces the parameter count in the subsequent fully connected layers and expands the global receptive field of the features, thereby enhancing the effective capture of contextual information within each feature channel. A fully connected layer then classifies the features

| Table 2 Detailed structural parameters of classification module. | |
|---|---|
| **Layers** | **Parameters** |
| Conv | C-256, K-3, S-2 |
| Dense | 128 |
| Dropout | 0.5 |
| Dense | 64 |
| Dropout | 0.5 |
| Dense | 32 |
| Dropout | 0.5 |
| Dense | 2 |

extracted by the preceding modules. To mitigate overfitting during training, a dropout rate of 0.5 is applied.

## EXPERIMENTAL SETUP

### Dataset and preprocessing

The synchronized ECG and PCG data used in this study were sourced from the PhysioNet/CinC Challenge 2016 dataset (PhysioNet2016) (*Liu et al., 2016*; *Goldberger et al., 2000*). This dataset comprises data collected from multiple institutions worldwide, categorized into subsets training-a through training-f based on their origin. This study utilized the training-a subset, which contains 409 records, including 405 pairs of synchronized ECG and PCG signals. These signals were recorded using a Welch Allyn Meditron electronic stethoscope (frequency response: 20 Hz–20 kHz) and resampled to 2,000 Hz. Of the 405 pairs, 117 were obtained from healthy subjects, and 288 were from subjects with cardiovascular diseases, including mitral valve prolapse, aortic disease, and other pathological conditions. A total of 17 pairs of records were manually excluded due to noise interference, which is inherent in data collection. Table 3 provides details of the training—a subset.

Because cardiac physiological state can vary between individual heartbeats, beat-to-beat segmentation was employed to capture the characteristics of each cardiac cycle accurately. Specifically, each record underwent Z-score normalization. Following normalization, the data were segmented and expanded based on the S1-to-S1 interval, using the provided individual heart sound annotations within the PCG signals. The S1 heart sound marks the closure of the mitral and tricuspid valves. It signifies the beginning of ventricular contraction's mechanical activity, occurring shortly after the R-wave in the synchronized ECG signal (*Li et al., 2020*; *Stodieck & Luttges, 1984*). Therefore, the S1-to-S1 interval represents a complete cardiac cycle, as illustrated in Fig. 6.

To ensure consistent input length for the deep learning model, segmented signal fragments were resampled on the time axis to a duration of 1 s. Table 4 provides details of the segmented dataset. The dataset does not provide information to determine whether different records originate from the same subject. Therefore, subject-specific division into training and testing sets was not feasible. Instead, the segmented data were divided into five

**Table 3 Train—a subset profile.**

| Type | Noisy | Clean | Sample rate | Mean duration (s) |
|---|---|---|---|---|
| Negative | 1 | 116 | 2,000 | 32.53 |
| Positive | 16 | 272 | 2,000 | 32.57 |

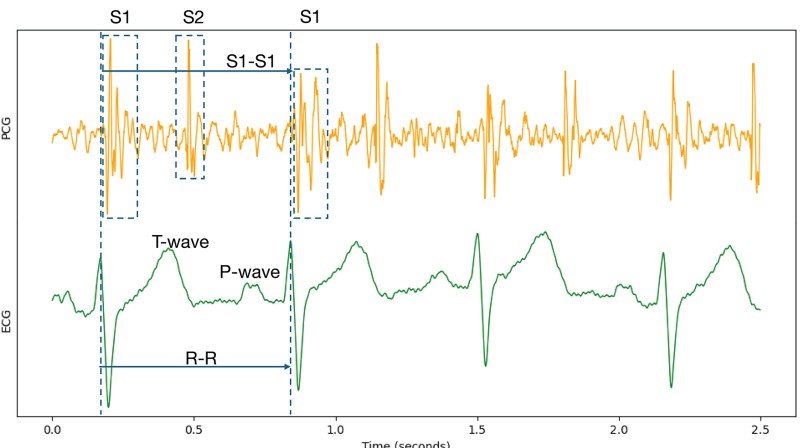

**Figure 6 Synchronized PCG and ECG signal waveforms.** T-wave and P-wave represent ventricular repolarization and atrial depolarization of ECG signals. S1 and S2 are the first and the second heart sounds of PCG signals.

**Table 4 Dataset profile after segmentation.**

| Type | Segments | Time duration (s) |
|---|---|---|
| Negative | 4,303 | 1 |
| Positive | 9,734 | 1 |

subsets for five-fold cross-validation. All experiments were conducted using this same data partitioning for training and testing.

## Model training environment

The experiments were conducted using the system equipped with an Intel 8255C CPU and two NVIDIA RTX 2080Ti GPUs. The software environment consisted of Python 3.8 and TensorFlow 2.9.0. Each model was trained for 100 epochs, utilizing the Adam optimizer and the cross-entropy loss function. The initial learning rate was set to 0.01. L2 regularization and dropout were employed to enhance generalization and prevent overfitting. A learning rate decay schedule was implemented: the learning rate was reduced by a factor of 0.1 if the training loss did not decrease for five consecutive epochs. Training was terminated if the training loss did not decrease for 20 consecutive epochs. Batch sizes of 32 and 128 were used during training and testing, respectively.

To address the class imbalance in the dataset, weight coefficients were applied to the positive and negative classes within the loss function. These coefficients were inversely proportional to the number of samples in each class. Furthermore, He initialization

(*He et al., 2015*) was applied to each layer of the model to accelerate training convergence and improve performance.

## Evaluation metrics

Five widely used evaluation metrics were employed to assess model performance: accuracy, sensitivity, specificity, area under the receiver operating characteristic curve (AUC-ROC), and F1-score. They are defined in Eqs. (4)–(8).

$$\text{Accuracy} = \frac{TP + TN}{TP + TN + FP + FN} \tag{4}$$

$$Sensitivity = \frac{TP}{TP + FN} \tag{5}$$

$$\text{Specificity} = \frac{TN}{TN + FP} \tag{6}$$

$$FPR = \frac{FP}{FP + TN} \tag{7}$$

$$F1 = \frac{2TP}{2TP + FP + FN} \tag{8}$$

where:

*TP* stands for true positive.

*FP* stands for false positive.

*TN* stands for true negative.

*FN* stands for false negative.

*FPR* denotes false positive rate.

## RESULTS AND DISCUSSION

To validate the effectiveness of the proposed PACFNet framework for classifying cardiac states using synchronized ECG and PCG signals, several experiments were conducted. These experiments compared the performance of different model configurations and analyzed the PACFNet approach against existing methods.

## Performance evaluation in missing modalities

To demonstrate the effectiveness of synchronized ECG and PCG multimodal signals for cardiac state classification, and to evaluate the PACFNet model's robustness in practical scenarios with missing modalities, the following experiments were conducted. The performance of single-modality branches (ECG-only and PCG-only) was compared to the performance of the full multimodal model. Additionally, the multimodal model's performance was assessed when either the ECG or PCG modality was absent. Table 5, Figs. 7 and 8 present the experimental results. For scenarios with a missing modality, the corresponding input values were set to zero, while the present modality remained unchanged.

The experimental results demonstrated several key findings. First, the proposed multimodal PACFNet model exhibited superior performance compared to single-modality

**Table 5 Performance comparisons between our proposed single-modality branch model and multimodal model methods.** Bold entries indicate the best performance for each metric.

| Model type | Accuracy | Specificity | Sensitivity | Precision | F1-score | AUC |
|---|---|---|---|---|---|---|
| Single_modal_ECG | 0.9615 | 0.9442 | 0.9691 | 0.9752 | 0.9721 | 0.9920 |
| Single_modal_PCG | 0.8179 | 0.7576 | 0.8446 | 0.8874 | 0.8655 | 0.8938 |
| Multi_modal_ECG | 0.9626 | 0.9421 | 0.9716 | 0.9743 | 0.9730 | 0.9928 |
| Multi_modal_PCG | 0.8312 | 0.8496 | 0.8231 | 0.9253 | 0.8712 | 0.9158 |
| Multi_modal_ECG_PCG | **0.9777** | **0.9728** | **0.9799** | **0.9879** | **0.9839** | **0.9967** |

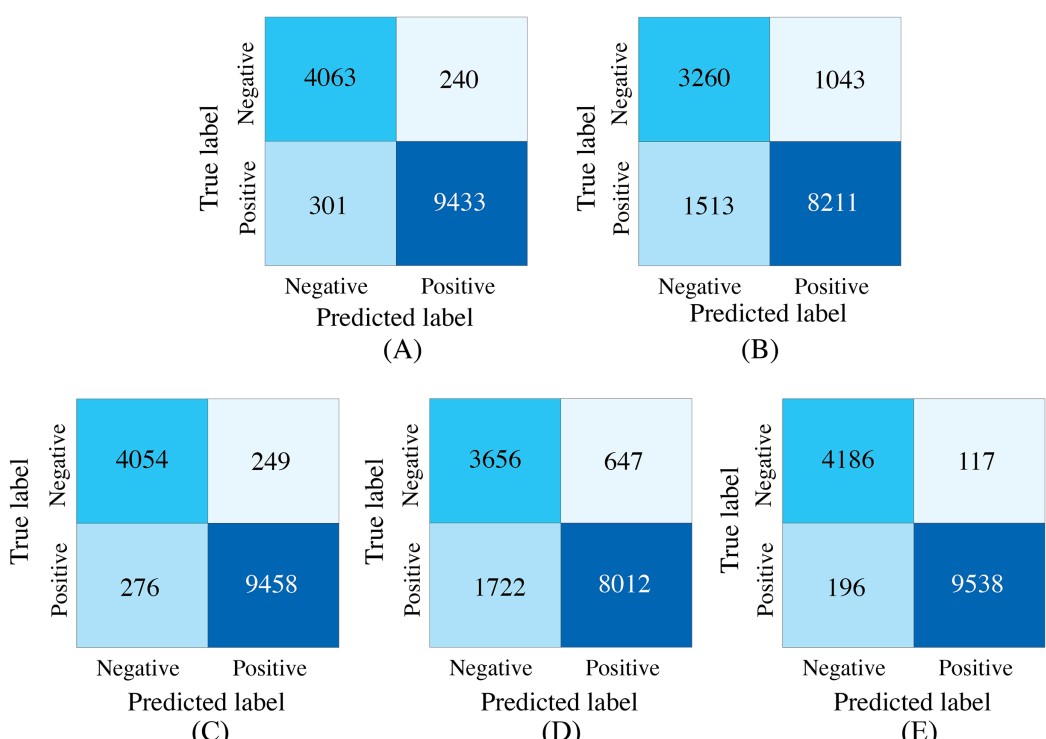

**Figure 7 Confusion matrices of our proposed single-modality branch model and multimodal model methods for the cardiovascular abnormality.** (A) ECG single-modality model. (B) PCG single-modality model. (C) ECG-only multi-modality model. (D) PCG-only multi-modality model. (E) Full model.

models when handling cases of missing modalities. Second, the highest performance was achieved when both ECG and PCG modalities were present. Specifically, for ECG signal classification, when the PCG modality was absent, the multimodal model achieved an average accuracy and AUC improvement of at least 0.11% and 0.08%, respectively, compared to the ECG-only model. With both ECG and PCG modalities present, the multimodal model showed improvements of at least 1.62%, 2.86%, 1.08%, and 0.47% in average accuracy, specificity, sensitivity, and AUC, respectively, compared to the ECG-only model. For PCG signal classification, when the ECG modality was absent, the multimodal model demonstrated an average accuracy and AUC improvement of at least 1.33% and 2.2%, respectively, compared to the PCG-only model.

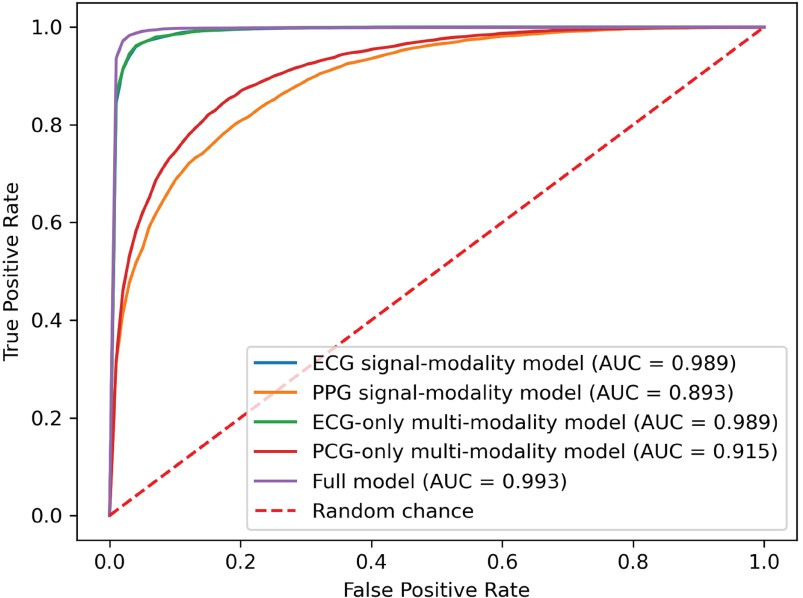

**Figure 8 ROC curves of our proposed single-modality branch model and multimodal model.**

These findings suggest that synchronized multimodal ECG and PCG signals provide complementary and richer pathological information for cardiac state classification, leading to improved accuracy. Furthermore, the proposed multimodal PACFNet model demonstrated superior performance compared to single-modality models even when one modality was absent. This is primarily attributed to the progressive multi-level feature fusion module within PACFNet. This module enhances important features and suppresses less relevant ones based on attention weights computed for each element of the feature vector. Consequently, the model maintains robust classification performance even with missing modality data.

## Performance evaluation of different feature fusion strategies

### Comparison of feature integration strategies in different stages

To further investigate the effectiveness of the proposed progressive feature fusion strategy, we compared PACFNet's performance with that of existing common fusion strategies in a multimodal setting. Additionally, to specifically evaluate the impact of the progressive feature fusion structure, we conducted comparative experiments using a single SACMF module applied after feature extraction from the individual ECG and PCG modality encoders. Table 6, Figs. 9 and 10 present the results of these comparisons.

Analysis of the comparison experiment results reveals that the proposed cross-modal fusion strategy, based on spatial and channel attention weights, outperforms existing common multimodal fusion approaches. Furthermore, the late fusion strategy exhibits superior performance compared to early fusion, with improvements of 0.62% and 0.4% in

Table 6 **Performance comparisons between our proposed SACMF module and common fusion strategies.** Bold entries indicate the best performance for each metric.

| Model type | Accuracy | Specificity | Sensitivity | Precision | F1-score | AUC |
|---|---|---|---|---|---|---|
| Early fusion | 0.9515 | 0.9328 | 0.9597 | 0.9700 | 0.9648 | 0.9882 |
| Late fusion | 0.9577 | 0.9551 | 0.9588 | 0.9797 | 0.9692 | 0.9922 |
| Only last SACMF | 0.9688 | 0.9668 | 0.9697 | 0.9851 | 0.9773 | 0.9936 |
| Full model | **0.9777** | **0.9728** | **0.9799** | **0.9879** | **0.9839** | **0.9967** |

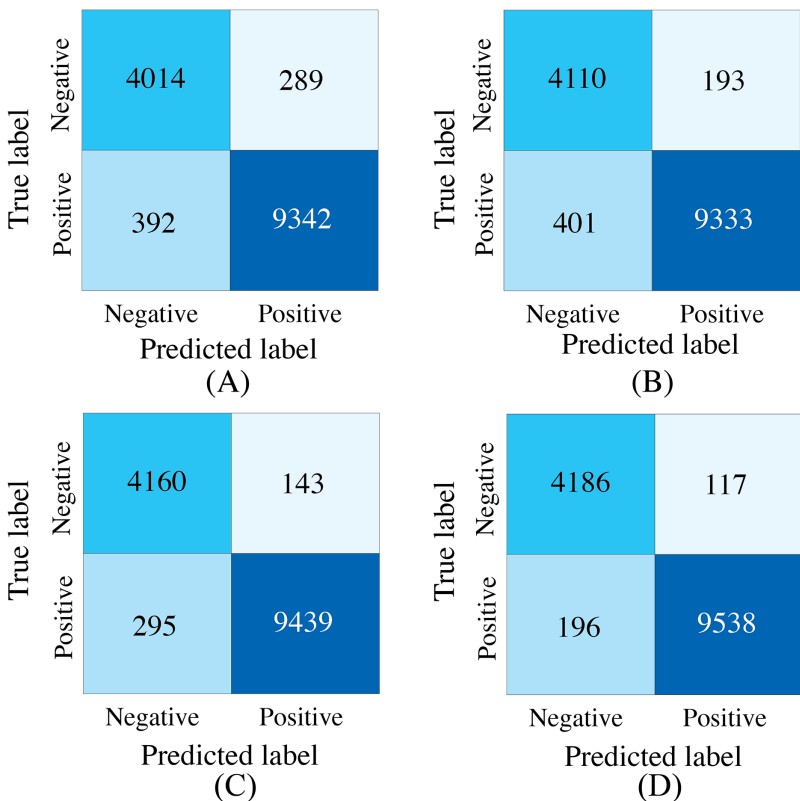

Figure 9 **Confusion matrices of our proposed SACMF module and common fusion strategies.** (A) Early fusion. (B) Late fusion. (C) Only last SACMF. (D) Full model.

average accuracy and AUC, respectively. Notably, specificity increased by 2.23% with late fusion, suggesting improved identification of negative samples.

Comparing the complete PACFNet model (with progressive fusion) to the model using only a single SACMF module at the late stage, we observe that progressive feature fusion achieves superior performance. Specifically, average accuracy increased by 0.89% with the progressive approach. This indicates that continuous multimodal feature fusion, progressing from shallower to deeper feature extraction levels, allows the model to learn more comprehensive information, thereby enhancing classification performance.

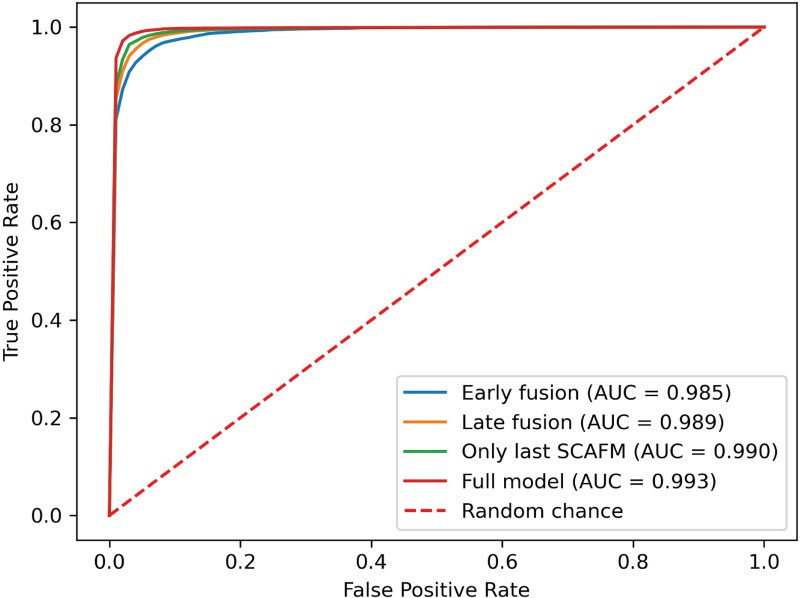

**Figure 10  ROC curves of SACMF module and common fusion strategies.**

### Comparison of different fusion strategies in identical backbone structures

To evaluate the effectiveness of the designed SACMF within the context of the PACFNet architecture, we conducted comparative experiments with different fusion module designs. These designs included: direct concatenation of features from the two modalities; only spatial attention weights; only channel attention weights; and our proposed complete SACMF module, incorporating both spatial and channel attention. Table 7, Figs. 11 and 12 present the results of these comparisons.

By analyzing the results of the comparison experiments, we can draw the following conclusions. Our proposed feature fusion strategy using both spatial and channel attention weights achieves better performance, outperforming other module designs in all evaluation metrics. This performance improvement primarily arises from the simultaneous utilization of channel and spatial attention mechanisms, which assess the significance of each positional element within the input feature vector. This mechanism enhances the model's sensitivity to critical information, thereby improving classification performance.

### Comparison with state-of-the-art methods

Table 8 compares the performance of our PACFNet model with that of state-of-the-art methods. The results demonstrate that PACFNet achieves superior classification performance.

*Li et al. (2022c)* considered both early and late fusion strategies in their multimodal approach. However, the resulting specificity was low. This observation aligns with our earlier findings, which indicated that direct concatenation of ECG and PCG signals at an early stage does not yield optimal classification performance, and that decision-level fusion in later stages offers limited improvement. Studies by *Li, Hu & Liu (2021)* and

**Table 7 Performance comparisons of different fusion module designs.** Bold entries indicate the best performance for each metric.

| Model type | Accuracy | Specificity | Sensitivity | Precision | F1-score | AUC |
|---|---|---|---|---|---|---|
| Concatenation | 0.9689 | 0.9626 | 0.9717 | 0.9833 | 0.9775 | 0.9941 |
| Only SA | 0.9631 | 0.9698 | 0.9601 | 0.9863 | 0.9730 | 0.9937 |
| Only CA | 0.9645 | 0.9628 | 0.9652 | 0.9833 | 0.9741 | 0.9940 |
| Full model | **0.9777** | **0.9728** | **0.9799** | **0.9879** | **0.9839** | **0.9967** |

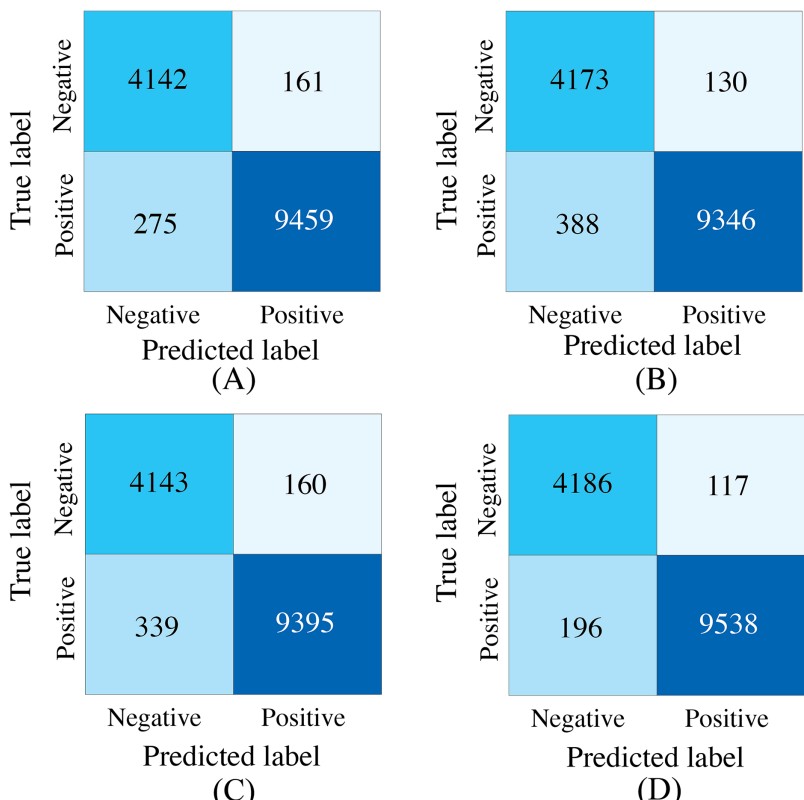

**Figure 11 Confusion matrices for different fusion module designs within the PACFNet architecture.** SA represents spatial attention, while CA denotes channel attention. (A) Simple concatenation. (B) Only SA. (C) Only CA. (D) Full model.

*Morshed & Fattah (2023)* utilized late fusion, extracting features from the ECG and PCG branches independently before fusing them for classification. The experimental results presented in the table suggest that this approach may not be sufficient to fully exploit the complementary information between the different modalities. In contrast, the approaches proposed by *Qi et al. (2023)*, *Zhang et al. (2024)*, and our PACFNet model perform cross-modal feature fusion during the feature extraction process. The work of *Qi et al. (2023)* transformed the signals into 2D images and input them into a Transformer model and a downsampling residual network for feature extraction and classification. *Zhang et al. (2024)* performed feature fusion during the progressive feature extraction process.
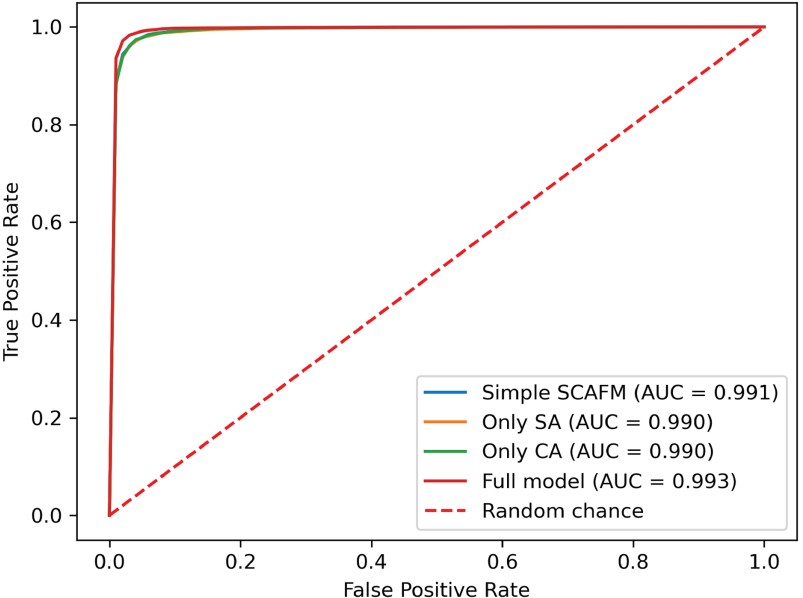

**Figure 12  ROC curves of different fusion module designs.**

**Table 8  Performance comparisons between our PACFNet and state-of-the-art methods.** Bold entries indicate the best performance for each metric.

| Method | Model | Accuracy | Specificity | Sensitivity | Precision | F1-score |
|---|---|---|---|---|---|---|
| *Li, Hu & Liu (2021)* | CNN+SVM | 0.936 | 0.845 | 0.903 | 0.874 | 0.873 |
| *Li et al. (2022c)* | BiLSTM+ GoogleNet | 0.961 | 0.908 | **0.985** | – | – |
| *Morshed & Fattah (2023)* | DNN | 0.951 | 0.909 | 0.951 | 0.95 | 0.99 |
| *Qi et al. (2023)* | Transformer | 0.943 | 0.909 | 0.977 | – | – |
| *Zhang et al. (2024)* | CNN | 0.944 | 0.939 | 0.948 | – | 0.973 |
| Proposed method | CNN | **0.977** | **0.973** | 0.98 | **0.984** | **0.997** |

The experimental results in Table 8 highlight the effectiveness of PACFNet's designed feature extraction and cross-modal fusion. Our modality-specific feature extraction, based on a powerful encoder, is capable of extracting multi-level features, progressing from superficial to deep representations of the input signal. Besides, our progressive cross-model feature fusion module, combining both spatial and channel attention mechanisms, can comprehensively analyze the contribution of each region across different levels of modal features. The PACFNet architecture effectively improves classification performance while maintaining a relatively small number of model parameters.

However, our proposed approach also has the following limitations:

(1) Due to the limitation of the dataset, the current study was conducted using a publicly available dataset (PhysioNet2016), and subsequent experiments in other private datasets are needed to further validate the classification performance of the model.

(2) The model's performance is contingent upon precise beat-to-beat segmentation of ECG and PCG signals, demanding highly accurate cardiac cycle annotations. Furthermore, the use of short data segments currently limits the model's capacity to account for inter-patient variability.

(3) Limitations in publicly available model architecture details and the lack of implementation code precluded a fair comparison of computational complexity. This aspect will be more thoroughly investigated in future work.

## CONCLUSION

In this study, we introduced PACFNet, an end-to-end deep learning model that significantly advances cardiac state detection by innovatively employing a progressive multi-level fusion strategy for synchronized ECG and PCG signals, which are pre-processed using a beat-to-beat segmentation approach to capture individual cardiac cycle dynamics. Our key contribution lies in the development of this novel architecture, featuring dedicated four-layer modality-specific encoders and, critically, the selective attention-based cross-modal fusion (SACMF) module. Unlike direct early fusion and late fusion approaches, SACMF utilizes cascaded spatial and channel attention mechanisms to dynamically weigh and select the most salient features from each modality at multiple hierarchical levels, enabling a comprehensive evaluation of feature importance. Evaluation on the PhysioNet 2016 dataset conclusively demonstrated PACFNet's superiority, as it not only outperformed current state-of-the-art multimodal methods in multimodal scenarios but also maintained remarkable robustness even with missing modalities. Therefore, PACFNet, leveraging beat-to-beat signal analysis and sophisticated attention-based multi-level fusion, offers a potent and effective solution for cardiac state determination, highlighting its significant potential in enhancing the accuracy and reliability of automated multimodal diagnostic systems.

In future work, we will focus on collecting a larger dataset of synchronized ECG and PCG signals from patients with diverse subtypes of heart disease and utilizing generative models to address the issue of class imbalance within the dataset. Building upon this enriched and balanced data foundation, we will develop more advanced models, emphasizing not only enhanced diagnostic accuracy for precise identification of different heart disease subtypes but also improved computational efficiency and reduced resource requirements. To rigorously evaluate the practical applicability of these models, we will conduct a comprehensive analysis and standardized benchmark comparing their computational complexity against relevant baseline and state-of-the-art methods.

### Funding

This work was supported by the Universiti Malaya Research Excellence Grant (Project Number: UMREG056-2024). The funders had no role in study design, data collection and analysis, decision to publish, or preparation of the manuscript.

### Grant Disclosures

The following grant information was disclosed by the authors:
Universiti Malaya Research Excellence Grant: UMREG056-2024.

### Competing Interests

The authors declare that they have no competing interests.

### Author Contributions

- Wei Peng Li conceived and designed the experiments, performed the experiments, analyzed the data, performed the computation work, prepared figures and/or tables, authored or reviewed drafts of the article, and approved the final draft.
- Joon Huang Chuah conceived and designed the experiments, authored or reviewed drafts of the article, and approved the final draft.
- Guo Jeng Tan analyzed the data, authored or reviewed drafts of the article, and approved the final draft.
- Chengyu Liu conceived and designed the experiments, analyzed the data, authored or reviewed drafts of the article, and approved the final draft.
- Hua-Nong Ting conceived and designed the experiments, analyzed the data, prepared figures and/or tables, authored or reviewed drafts of the article, and approved the final draft.

### Data Availability

The Heart Sound Recordings dataset is available at: https://physionet.org/content/challenge-2016/1.0.0.

The code is available at GitHub and Zenodo:

- https://github.com/lightyLi/PACFNet-for-CVDs-detection.
- lighty. (2025). lightyLi/PACFNet-for-CVDs-detection: update_readme (update_readme). Zenodo. https://doi.org/10.5281/zenodo.15450169.

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
