# Peer review of "A progressive attention-based cross-modal fusion network for cardiovascular disease detection using synchronized electrocardiogram and phonocardiogram signals"

_PeerJ Computer Science, doi:10.7717/peerj-cs.3038_

## Round 0.1 · original submission · Major Revisions

Dear Authors,

Your paper has been reviewed. It needs major revisions before being accepted for publication in PEERJ Computational Science. More precisely

1) You should test your model with your dataset. Furthermore, you must add a complexity comparison using cross-validation. To validate your results, you must add a graphical analysis, such as a histogram, intensity profile, correlation heatmap, or spider graph analysis.

2) You must rewrite:

 a) The methods section: it could benefit from further elaboration, especially on how the attention modules function within the fusion process. You should expand the ablation study to include models trained with only ECG or PCG inputs to assess the individual contribution of each modality.

b) The conclusion section: You must add concluding and contribution points briefly. The abstract needs to be rewritten to better summarize the paper.

Reviewer 1 ·

Basic reporting

-

Experimental design

-

Validity of the findings

-

Additional comments

Explain how the present manuscript differs from and improves upon these established previous works. Include a study with your survey papers, and be sure to highlight any knowledge gaps that have come up recently.

Check and if in case missing, discuss some major sections in the article like the real-time/ practical application of the paper, future perspectives, major contribution in the paper, and significance of this study.

In the introduction section, include some other research applications as literature. Some suggestions are: Multi-modal medical image fusion framework using co-occurrence filter and local extrema in NSST domain; An end-to-end content-aware generative adversarial network-based method for multimodal medical image fusion; TSJNet: A Multi-modality Target and Semantic Awareness Joint-driven Image Fusion Network; Clustering-based Multi-modality Medical Image Fusion; Directive clustering contrast-based multi-modality medical image fusion for smart healthcare system.

The authors should test their model with their dataset (not only from the public dataset). The complexity comparison should be made. Cross-validation is lacking in this paper.

For the validation and best performance of the result, use graphical analysis like a histogram, intensity profile, correlation heatmap, or spider graph analysis.

Rewrite the conclusion. Add concluding, contribution points in it briefly. Abstract needs to be rewritten with a better summary of the paper.

Reviewer 2 ·

Basic reporting

The manuscript is written in clear and professional English throughout. The introduction provides sufficient context for understanding the importance of cardiovascular disease detection using physiological signals. However, some sections, particularly the methodological architecture description, are dense and hard to follow. A simplified explanation or an enhanced diagram would significantly improve clarity.

The article is professionally structured with appropriate figures and tables. However, there is no mention of code availability or preprocessing scripts, which limits reproducibility. I strongly recommend that the authors provide a link to the code and data processing pipeline to meet modern expectations for open research.

Experimental design

The study addresses an important and timely research question: enhancing cardiovascular disease detection using synchronized ECG and PCG signals. The proposed PACFNet model is original and aligns well with the journal’s scope. The study is methodologically rigorous, and the experimental evaluation on the PhysioNet 2016 dataset is appropriate.

However, the methods section could benefit from further elaboration, especially on how the attention modules function within the fusion process. Moreover, the ablation study should be expanded to include models trained with only ECG or only PCG inputs to assess the individual contribution of each modality.

Validity of the findings

The results presented are statistically sound and convincingly support the authors’ claims. The performance metrics on the dataset are impressive. Nevertheless, the lack of code availability raises questions about reproducibility, and the contribution of each signal modality remains partially unexplored due to the limited ablation study.

The conclusions are appropriate and aligned with the research question, but their generalizability would be strengthened with further analysis, such as comparing unimodal vs. multimodal inputs.

Additional comments

To contextualize the contribution of ECG-based diagnostic methods, I recommend citing the following relevant work:
A novel ternary pattern-based automatic psychiatric disorders classification using ECG signals — this paper highlights the broader potential of physiological signals in AI-driven healthcare applications and complements the direction of the current study.

·

Basic reporting

The paper was written clearly, however, there are a few fragments that the authors should address:
Introduction
- Line 50: The first sentence, "Cardiovascular diseases (CVDs) are a significant global health concern," could be paraphrased as "Cardiovascular diseases (CVDs) are a major concern for global health."

Literature review
-Line 116: notable → remarkable

Dataset and preprocessing, and Table 3
- "Train-a" to "Train-f" → "Training-a" to "Training-f". The analyzed dataset uses names "training-a" to "training-f".

The references and background are written adequately, however, lines 67-75 could be paraphrased to emphasize the objective and the background. The use of "This method should utilize" and the following sentences suggest a research grant proposal, not a manuscript of an original research article.

The rest is presented clearly and comprehensively.

Experimental design

The experiment was described clearly and thoroughly, however, I noticed a misspelling of the subsets in the analyzed dataset in the Dataset and preprocessing section and Table 3:
"Train-a" to "Train-f" should be rewritten as "Training-a" to "Training-f" for each mentioned training subset. The analyzed dataset uses names "training-a" to "training-f".

Validity of the findings

The findings have been clearly and comprehensively presented and explained.

---

## Round 0.2 · Minor Revisions

Dear Authors,

Your paper has been revised. It needs minor revision before being accepted for publication in PEERJ Computer Science. More precisely:

1) All figures contain lossy compression artifacts that can be seen behind the text. This problem is more visible in Figures 1 and 2 than in Figures 3-5. Therefore, you must replace the figures with versions of higher resolution and/or a higher DPI (dots per inch).

2) When referring to literature references, use space before reference(s) across the manuscript.

Reviewer 2 ·

Basic reporting

The authors have completely addressed all my comments, and I have no further concerns. Therefore, I recommend accepting the paper.

Experimental design

The authors have completely addressed all my comments, and I have no further concerns. Therefore, I recommend accepting the paper.

Validity of the findings

The authors have completely addressed all my comments, and I have no further concerns. Therefore, I recommend accepting the paper.

Additional comments

The authors have completely addressed all my comments, and I have no further concerns. Therefore, I recommend accepting the paper.

·

Basic reporting

The authors have addressed reviewer comments and applied them to revise the manuscript. The literature references are adequate, the structure, table and figure design are accurate. However, the authors should eliminate lossy compression artifacts in figures. And when referring to literature references, the authors should use a space before reference(s) across the manuscript.

Experimental design

The research question has been clearly defined and remains meaningful. The experiment has been comprehensively described.

Validity of the findings

The data have been analyzed accordingly and the conclusions link to the research question and experimental results.

Additional comments

The authors have addressed reviewer comments and applied them to improve the manuscript to make it more acceptable for publishing. However, I noticed three minor points that should be addressed:
1. When referring to literature references, use space before reference(s) across the manuscript.
2. Line 69: "has fraught with" could be replaced by "has".
3. All figures contain lossy compression artifacts that can be seen behind the text. This problem is more visible in Figures 1 and 2 than Figures 3-5. Therefore, I recommend replacing the figures with versions of a higher resolution and/or number of DPI (dots per inch).

---

## Round 0.3 · accepted · Accept

Dear Authors,

Your paper has been revised. It has been accepted for publication in PEERJ Computer Science. Thank you for your fine contribution.

·

Basic reporting

I am satisfied with the revisions and have no further comments.

Experimental design

no comment

Validity of the findings

no comment

Additional comments

No further comments.